# Synthesis of Triblock Polycarboxylate Superplasticizers with Well-Defined Structure and Its Dispersing Performance in β-Hemihydrate Gypsum

**DOI:** 10.3390/molecules28020513

**Published:** 2023-01-04

**Authors:** Guangming Guo, Guohua Gao, Weiliang Jiang, Xianglong Wang, Meishan Pei, Luyan Wang

**Affiliations:** 1Shandong Hi-Speed Zilin Expressway Co., Ltd., Zibo 255100, China; 2Shandong Hi-Speed Engineering Test Co., Ltd., Jinan 250003, China; 3School of Chemistry and Chemical Engineering, University of Jinan, Jinan 250022, China

**Keywords:** gypsum, hemihydrate, triblock, polycarboxylate, superplasticizer, dispersion

## Abstract

In this work, a novel A_a_BA_b_-type triblock polycarboxylate superplasticizers (PCEs) with well defined molecular structures were designed and synthesized, firstly, by reversible addition-fragmentation chain transfer (RAFT) polymerization, to explore the structure–property relationship PCEs in the β-hemihydrate gypsum (β-HH) system. Three PCEs with the same molecular weight and different structure were obtained by changing the feed ratio of the RAFT agent, initiator, and monomer. The effect of the chemical structure of PCEs on their dispersing property and water reduction capacity were assessed in gypsum by measuring the flowability of pastes and the adsorption ability of PCEs on gypsum. Results showed that among three PCEs, when the monomer ratio is 5:1 and a:b = 1:1, PCE-1 exhibited a higher working efficiency, verifying the contribution of regulating structural parameters to the improvement in performances of gypsum paste, because PCE-1 showed the strongest binding capacity with calcium ions due to the relatively equal amount of carboxyl groups at both ends. The A_a_BA_b_-type PCEs provide a special advantage over the conventional comb polymer to understand the relation between the structure and property of PCEs, and a direction for further development of PCEs of high performance.

## 1. Introduction

In recent years, non-renewable energy consumption has been increasing with the rapid development of society, and energy preservation has become a global issue. The importance of sustainable development and natural resource preservation has been recognized by various industries [1,2,3,4]. In the construction industry, cement is one of the most important building materials. However, the energy consumption of cement clinker sintering is relatively large and the emissions of CO_2_ during cement production increase global warming [5]. Significantly, the energy consumption in the production of gypsum, which is another important cementitious material in construction, is much lower than that of cement. Moreover, besides the abundant natural gypsum reserves, a huge amount of gypsum has been emitted by industries as by-products, with a potential danger to the environment. The efficient use of by-product gypsum, which is usually utilized to produce building gypsum, will help protect the environment and realize the recycling of waste resources.

The building gypsum is mainly composed of β-hemihydrate gypsum (β-HH). β-HH is widely applied throughout the world due to its easy manufacture, aesthetics, low price and environmental friendliness [6,7]. In addition, it has fine grains, a large specific surface area, and a large level of water consumption for its standard consistency. The theoretical water consumption for gypsum hydration is 18.6% of gypsum quality. However, in order to maintain the plasticity of gypsum slurry in the actual application, the added water accounts for 60–80% of the mass of gypsum. After the gypsum is hardened, the hardened body always exhibits a high porosity and reduction in strength. Such a problem also exists in cement applications, wherein polycarboxylate superplasticizers (PCs) are generally used to solve this problem of the high water demand in cement. 

PCs have been largely applied to the field of construction since its invention [8,9,10,11,12,13,14]. PCs are formed by the main chain with carboxylate groups and the polyethylene oxide segment side chain, exhibiting excellent water-reducing properties and a high compatibility and processability in cement-based material; nowadays PCs are accepted as the most efficient dispersant [15,16,17]. Similarly, they have shown excellent ability in dispersing the gypsum system in practice, and the water reduction mechanism of PCs can also be applied to gypsum [18,19,20,21,22]. When β-HH is hydrated, the surface of gypsum particles is positive due to the positive electricity of calcium ions. When PC is added to the gypsum paste, the carboxylic acid groups in the main chain can complex with the calcium ions on the surface of gypsum through electrostatic interaction and adsorb on the surface of gypsum particles. Concurrently, the polyethylene oxide side chains can provide the steric hindrance to hinder the aggregation of gypsum particles. Both interactions mentioned above between PCs and gypsum help improve the dispersing properties of gypsum and reduce the water demanded in the work process of gypsum. 

At present, studies on the water reducers in gypsum are fewer than that in cements [18,19,20,21,22]. In particular, the relationship between the structure of PCs and their property in gypsum need detailed investigation because the efficiency of PCs is obviously dependent on their molecular structure [23]. Nevertheless, PCs are generally random copolymer synthesized by the free radical aqueous solution polymerization method; the monomers are randomly arranged in the macromolecular structure, which is inconvenient to explore the structure–property relation of PCs [24,25,26]. 

In this work, a novel type of triblock polycarboxylate superplasticizers (PCEs) with well-defined molecular structure were synthesized by reversible addition-fragmentation chain transfer (RAFT) polymerization and used as the gypsum water reducer for the first time. The chemical structure of PCEs was analyzed by Fourier transform infrared spectroscopy (FTIR), nuclear magnetic resonance spectroscopy (^1^HNMR), and gel permeation chromatography (GPC). The effect of the chemical structure of PCEs on their performance in β-HH was investigated by measuring the flowability, adsorption ability, and hydration of β-HH doped with PCEs. This research is expected to offer a new and efficient way to explore precisely the relationship between the PCEs structure and their performance in gypsum.

## 2. Results

### 2.1. Structure Characterization of PCEs

FTIR spectra of tBA, PtBA, TPEG, PtBA-PTPEG and PtBA-PTPEG-PtBA are shown in Figure 1. In the spectrum of tBA, two recognizable peaks at 1725 cm^−1^ and 1630 cm^−1^ were attributed to carbonyl group (–C=O) and unsaturated double bond (–C=C–). In the spectrum of PtBA, the adsorption peaks of the carbon–carbon were disappeared, which confirmed the successful preparation of PtBA sample by RAFT polymerization. In the spectrum of TPEG, the stretching vibrations of ether linkage (–C–O–C–) were evidenced by the peak at 1110 cm^−1^. As seen in Figure 1, the spectra of PtBA-PTPEG and PtBA-PTPEG-PtBA contained functional groups such as ether group –C–O–C– and carboxyl group –C=O, which confirmed the successful preparation of PCEs sample by RAFT polymerization. The spectra of FTIR verify the introduction of characteristic groups into the macromolecules, which were in accordance with the expected structure.

Figure 2 shows ^1^HNMR spectra of TPEG, PtBA, PtBA-PTPEG and PtBA-PTPEG-PtBA. The signal at δ = 1.5 ppm in curve of PtBA was characteristic of tBA. Curve of TPEG showed the signal of –CH_2_–CH_2_–O– at δ = 3.50–3.65 ppm. Curves of PtBA-PTPEG and PtBA-PTPEG-PtBA also have the signals of δ = 1.5 ppm and δ = 3.50–3.65 ppm. At the same time, the acid-to-ether ratio, obtained by calculation after integrating the peak area on the curve at δ = 1.5 ppm and δ = 3.50–3.65 ppm, is in line with expectations. The 1HNMR spectra of Figure 2c,d represent PtBA-PTPEG and PtBA-PTPEG-PtBA were synthesized successfully, because another new peak appeared besides TPEG. Results verify the introduction of characteristic groups into the as-synthesized macromolecules, which were in accordance with the designed structure.

In this work, three PCEs with the same molecular weight and different monomer ratios were synthesized by changing the feed ratio of RAFT agent CTA and initiator AIBN. The obtained molecular weight listed in Table 1 confirmed that the PCEs synthesized by RAFT polymerization are consistent with the expected molecular weight.

### 2.2. Dispersion Performance of PCEs

The fluidity of β-HH represents the dispersion ability of PCEs in β-HH plaster. The experimental results of PCE-1 at dosage 0.1% to 1% (W/H = 0.65) are illustrated in Figure 3. It can be concluded that the fluidity of gypsum samples added with PCEs are higher than that of the blank sample. It can also be seen from Figure 3 that the fluidity of gypsum plaster is related to the content of PCEs under normal consistency water consumption. As the dosage of PCEs increases, the fluidity of β-hemihydrate gypsum samples grow up gradually. As the PCEs content is 0.4%, the fluidity of gypsum plaster reaches 298 mm, a higher dispersion efficiency. When the PCE dosage continues to increase, the fluidity is almost unchanged.

Based on this, PCE-2 and PCE-3 were also added to β-HH system (W/H = 0.65) at a dosage of 0.4%, respectively, and the initial fluidity of gypsum plaster was tested and shown in Figure 4.

In Figure 4, the initial fluidity of β-HH gypsum plaster with PCEs are obviously better than that of the blank sample. The fluidity values of PCE-2 and PCE-3 are also lower than that of PCE-1. Compared with PCE-1, the ratio of AA/TPEG in PCE-2 and PCE-3 and the feeding ratio of AIBN and CTA for these two PCEs are the same, but the ratios of carboxyl groups at both ends of polymer molecules are different. In the early hydration of β-HH, the gypsum particles tend to encapsulate a large number of water molecules so that massive water needs to be added to achieve working fluidity. The high water demand will lead to the strength reduction. The application of PCEs in gypsum to reduce water consumption and increase the strength of gypsum products is of great significance. For PCEs, the anionic carboxylate group located on the polymer backbone has a strong chelating interaction with Ca^2+^, which help the negatively charged PCE molecules be adsorbed on β-HH particles [27,28]. Subsequently, the polyether side chains of PCE, based on the steric repulsion mechanism, can disperse β-HH particles away from each other and release water molecules to improve the plasticity of gypsum [24]. The calcium binding capability of PCEs depends on the number of carboxylate groups and their steric position along the polymer backbone. Thus, the polymer architecture plays an important role in the effective anionic charges of PCEs in β-HH system [27]. PCE-1, with the ratio of carboxyl groups at both ends of polymer molecules being 1:1, could be adsorbed on β-HH particles as much as possible. PCE-2 and PCE-3 with the ratio of carboxyl groups at both ends of polymer molecules are different. The number of carboxylate groups and their steric along the polymer trunk may affect the effective anion charges of PCEs molecules adsorbed on β-HH particles. The reduction of effective anion charges leads to a reduction of PCEs adsorbed on β-HH particles, which results in the decrease in the steric hindrance effect of PCE-2 and PCE-3, and thus β-HH particles tends to cluster together. This may further be corroborated by the adsorption curves shown in Figure 5.

The adsorption behaviors of PCEs on β-HH is one of the most effective methods to detect β-HH interaction with PCEs, similar to the cement system [29,30]. When PCEs were added to β-HH system, firstly, the molecules should be adsorbed on β-HH particles by carboxyl groups, and then the side chains will disperse particles by steric hindrance effect. 

The adsorption curves of PCE-1 at a dosage of 0.2% to 0.5% in β-HH plaster (W/H = 10) are displayed in Figure 5a. The adsorption amount on the β-HH particles increased with time until the stage of adsorption saturation was obtained with PCE-1 dosage of 0.4%. This is inconsistent with the result of Figure 3, wherein the fluidity of gypsum plaster does not increase with the amount of PCE-1 in the gypsum, when PCE-1 dosage was 0.4%. The adsorption amounts of PCE-1, PCE-2, and PCE-3 at the dosage of 0.4% in β-HH plaster are displayed in Figure 5b, respectively. As can be seen, PCE-1 is easily absorbed on gypsum particles, and the adsorption ability of PCE-2 and PCE-3 decreased sequentially. Therefore, at the same dosage, the polymer molecule with more similar numbers of carboxyl groups at both ends are more conducive to adsorption on gypsum particles. It could be inferred that the molecular structure and configuration of PCEs are largely responsible for the adsorption difference [31].

In order to investigate further the influence of the different structures of PCEs on their interaction with calcium ions, the electrical conductivity of calcium chloride (CaCl_2_) solution at a mass concentration of 5% and the conductivity after adding PCEs are measured, respectively, as shown in Figure 6.

It can be seen that the conductivity of the CaCl_2_ solution is 71.5 ms/cm. When PCE is added, the conductivity of the solution decreases, wherein that of the sample with PCE-1 decreases the most. This shows obviously that PCE-1 binds to more Ca^2+^. The calcium binding capability of PCEs depends on the number of carboxylate groups and their steric position along the polymer trunk. The carboxyl group in PCE-1 has a stronger binding capacity with calcium ions due to the relatively equal amount of carboxyl groups at both ends. In PCEs with high side chain density and long side chain, the accessibility of the carboxyl group ligands may be stereochemically restrained. Due to steric interactions, as shown in Figure 7, calcium ions bind to carboxylic acid groups in a bidentate manner [32]. From the perspective of coordination chemistry, the more symmetrical the multiple coordination groups in PCEs, the smaller the steric hindrance, and the smaller the driving energy required to generate chelate molecules when PCEs coordinated with Ca^2+^. Therefore, the distribution of carboxyl groups in PCEs molecule are more symmetrical, and it is easier for the carboxyl groups to chelate with gypsum particles. In addition, polycarboxylates can form calcium chelate complexes of various forms. Ca^2+^ complexation in PCEs may occur through carboxyl coordination in polymer molecules or carboxyl coordination between polymer molecules. When PCE-2 and PCE-3 form chelate complexes with Ca^2+^, part of the carboxyl groups are not complexed with Ca^2+^ due to the difference in the number of carboxyl groups at two ends. The dispersion of PCE-1 is more than PCE-2 and PCE-3. The effect of the arrangement of the carboxyl groups of PCE molecule on the complexing ability of Ca^2+^ deserves further investigation in the future.

### 2.3. Heat Flow Calorimetry

As shown in Figure 8, the PCEs obviously decelerate the hydration of β-HH and also decrease the height of the hydration peak when compared to the blank sample. As is consistent with results reported, PCEs have the potential to delay gypsum hydration [25]. Evidently, the effect of PCE-1 on gypsum hydration is more significant because the heat flow peak appeared at about 70 min which is much longer than that of other samples.

### 2.4. SEM

The microstructure of hardened gypsum was observed by SEM, as shown in Figure 9. In the blank gypsum system without PCE-1, the crystals of the hydration product showed needle-like structure and interlocked each other. When PCE-1 was added, SEM images confirmed that the hardened body of β-HH admixed with PCE-1 exhibited obviously larger crystals of gypsum, at a microscopic level, than that from the blank. This is in accordance with the result shown in Figure 8, to a certain extent.

### 2.5. Mechanical Properties

The strength of the hardened gypsum from β-HH is affected by the amount of water mixed to β-HH. The water consumption for specimens was adjusted to achieve the gypsum paste flow at 180 ± 5 mm. Then, the compressive strength and flexural strength of the hardened body of β-HH mixed with PCEs at a dosage of 0.4% were tested after they were cured for 7 d. The results are shown in Table 2.

The flexural and compressive strength of specimens with PCEs is significantly higher than that of the blank sample. When the same fluidity is achieved, the amount of water used for achieving this fluidity is reduced when PCEs were introduced into β-HH specimens. In addition, compared with samples containing PCEs, PCE-1 doped sample exhibited the highest flexural and compressive strength. It can be attributed to the excellent dispersion ability of PCE-1 in β-hemihydrate gypsum.

## 3. Materials

### 3.1. Materials

The gypsum in the experiments was provided by Shandong Huize New Building Materials Co., Ltd (Jinan, China). The isoprenyl oxy polyethylene glycol ether (TPEG, with *M*_w_ = 2400) was obtained from Shandong Zhuoxing Chemical Co., Ltd (Wudi, China). The monomer tert-butyl acrylate (tBA) with analytical grade was obtained from Shanghai Maclin Biochemistry Technology Co., Ltd (Shanghai, China). and used without further purification. The initiator 2,2′-Azobis (2-methylpropionitrile) (AIBN) was supplied by Shanghai Aladdin Reagent (Shanghai, China), and it was recrystallized twice from methanol before use. Also supplied by Shanghai Aladdin Reagent (Shanghai, China), 2-dodecylsulfanylcarbothioylsulfanyl-propanoic acid (CTA) as the chain transfer agent was prepared in the laboratory.

### 3.2. RAFT Polymerization of PCEs

In this work, tBA was used as monomer A and polyether macromonomers TPEG was used as monomer B. A triblock polymers were first synthesized from A and B via RAFT polymerization, and then hydrolyzed to be polycarboxylate superplasticizers (PCEs) with A_a_BA_b_ structure. To investigate the effect of structural parameter on the dispersion capacity of β-HH, the PCEs with different structures were obtained by changing the feed ratio of RAFT agent CTA, initiator AIBN, and monomer. All the experiments in this study are shown in Table 3.

### 3.3. Measurement

The PCEs specimens used for all the characterizations and measurements were placed in a dialysis bag (MWCO 20,000 or 40,000), in which the water was changed every 3–5 h lasting for 2 days to remove the unconverted reactants, and then dried by freeze-drying.

#### 3.3.1. Fourier Transform Infrared Spectroscopy (FTIR)

The absorption spectrum was tested by FTIR spectrometer (Bruker Vertex 70, saarbrucken, Germany). The samples were prepared with potassium bromide (KBr) pellets and the spectrum of PtBA, TPEG, and PtBA-PTPEG-PtBA were recorded at a resolution of 4 cm^−1^.

#### 3.3.2. ^1^H Nuclear Magnetic Resonance Spectroscopy (^1^HNMR)

^1^HNMR spectra of PtBA was obtained by an Avance III 400 MHz NMR spectrometer (Bruker, Faellanden, Switzerland) in deuterium acetone with tetramethyl silane as the internal standard at room temperature. ^1^HNMR spectra of TPEG, PtBA-PTPEG, and PtBA-PTPEG-PtBA were obtained in deuterium oxide with tetramethyl silane as the internal standard at room temperature.

#### 3.3.3. Gel Permeation Chromatography

The PtBA was determined using a Waters 1500 gel chromatography (Milford, MA, USA) with tetrahydrofuran as the eluent at a flow rate of 1.0 mL/min. PtBA-PTPEG and PtBA-PTPEG-PtBA were hydrolyzed and characterized by PL-GPC50 gel chromatography at a flow rate of 1.0 mL/min. The concentration of the sample solution is 5 mg/mL, and all sample solutions were filtered with a 0.45µm filter membrane prior to the test.

#### 3.3.4. Fluidity of Gypsum Plaster

The water requirement for β-HH normal consistency was measured according to Chinese standard GB/T17669.4-1999, “gypsum plasters determination of physical properties of pure paste”, by the Standardization Administration of the People’s Republic of China. When the slump flow of β-HH reaches 180±5 mm, the water–gypsum ratio (W/H) for standard consistency is recorded. In this work, the fluidity of β-HH plaster with different content of PCEs was determined at the same water–gypsum ratio. Two-hundred grams of β-HH and PCE with a certain dosage (weight percent of PCEs content in β-HH) are mixed evenly in 5 s with standard consistency water quantity. The uniform plaster was acquired after stirring for 30 s. Then, it was immediately poured into a mini-slump cone (the height of 60 mm, the upper diameter of 36 mm, and the bottom diameter of 60 mm). The mini-slump cone is then removed vertically, and the average diffusion diameter of the plaster is measured. Each sample was tested three times, and the average value was calculated as the final fluidity value. Under the same standard consistency and water consumption, testing the fluidity of gypsum slurries with different contents of PCEs can characterize the dispersion properties of different PCEs in gypsum.

#### 3.3.5. Adsorption Measurement

The adsorption capacity of PCEs samples on β-HH was determined by the depletion method. Different β-HH samples (W/H = 10) were made at various PCEs dosages. Gypsum paste was centrifuged at 8500 rpm for 7 min to obtain the supernatant. Then the supernatant was filtered with a 0.45μm membrane. Organic carbon content was measured by the total organic carbon survey meter (TOC, Shimadzu TOC-5000A, Kyoto, Japan). According to the change of PCEs concentration in the aqueous phase, before and after the PCEs solution was incorporated into gypsum, the adsorption amount of PCEs on the surface of β-HH could be calculated.

#### 3.3.6. Heat Flow Calorimetry

Impact of the PCEs samples on gypsum hydration was captured at 25 °C by isothermal heat flow calorimetry (TAM Air). In a glass ampoule, 1 g of water was added to 2 g of gypsum containing PCEs with a dosage of 0.1%. The heat flow of the sample within 24 h was recorded after stirring.

#### 3.3.7. Morphologycharacterization

The scanning electron microscope (SEM, Gemini300, Oberko Heinz, Germany) was used to observe the fracture morphology of the hydrate gypsum product after curing for 7 days. In this measurement, the sample was coated with a gold coating to improve conductivity and prevent charging before scanning.

#### 3.3.8. Strength

The effects of PCEs on mechanical properties of β-HH were tested according to Chinese Standard (GB/T 17669.3-1999). The specimens were prepared with β-HH and PCEs with the dosage of 0.4%. The β-HH specimens maintain the same gypsum paste flow. Then, they were molded at 40 mm × 40 mm × 160 mm, and demolded after 24 h. Subsequently, they were unceasingly cured at this condition for the required ages maintained in water tanks at 20 °C. Three prismatic specimens were tested at 7 d hydration age, and values of compressive strength and flexural strength were obtained.

## 4. Conclusions

In this paper, a new kind of A_a_BA_b_-type triblock polycarboxylate superplasticizers with well-designed molecular structures were synthesized by RAFT polymerization. When PCEs were applied to β-HH system, PCE-1 with a considerable proportion of carboxyl groups at both ends of the polymer molecule can exert the greatest dispersibility in gypsum, and the hardened gypsum containing PCE-1 exhibit a higher flexural and compressive strength than that of other samples. From data of flowability, adsorption, hydration and electrical conductivity, the effect of the arrangement of the carboxyl groups of PCE molecule on their complexing ability with Ca^2+^ and their steric hindrance between gypsum particles shows that PCE-1 presents the strongest binding with Ca^2+^, of the three PCEs. Therefore, it deserves detailed investigation in the future to explore the relationship between structure parameters of A_a_BA_b_-type PCEs and their property. In contrary to PCE random copolymers reported in the previous literature, this work demonstrated that PCEs with a well-designed structure could be utilized as a satisfactory model to reveal in detail the effect of PCEs on gypsum system. As a green building material, β-HH will be used more and more widely; therefore, improving the application performance of gypsum by exploring the structure–property relation of PCEs is of great significance to sustainable development.

## Figures and Tables

**Figure 1 molecules-28-00513-f001:**
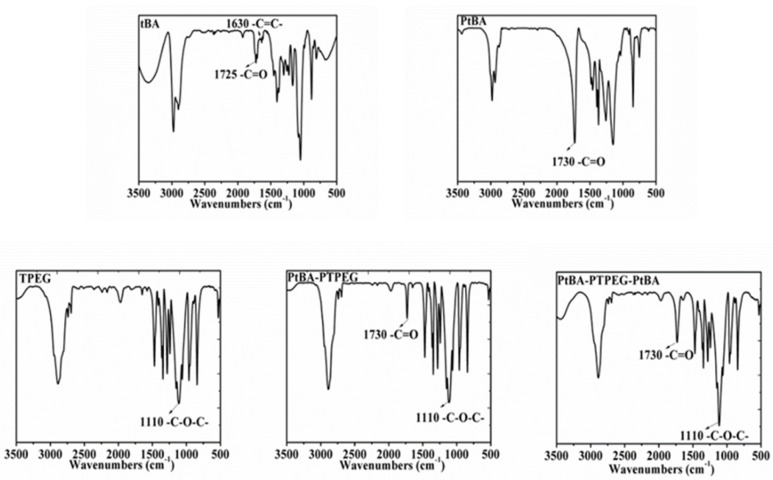
FTIR spectra of tBA, PtBA, TPEG, PtBA-PTPEG and PtBA-PTPEG-PtBA.

**Figure 2 molecules-28-00513-f002:**
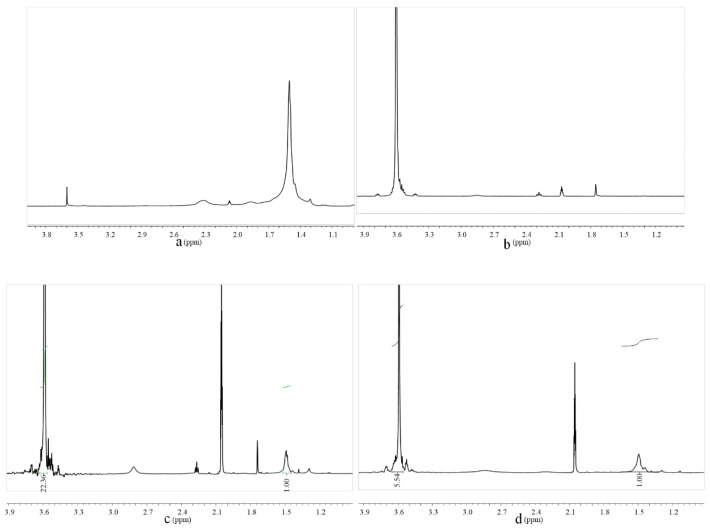
^1^HNMR spectrum of PtBA (**a**), TPEG (**b**), PtBA-PTPEG, (**c**) and PtBA-PTPEG-PtBA (**d**), respectively.

**Figure 3 molecules-28-00513-f003:**
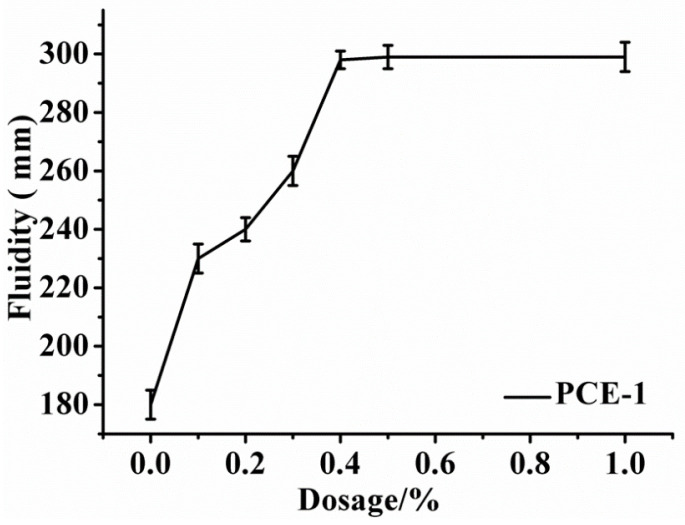
Initial fluidity of β-HH plaster (W/H = 0.65) admixed with PCE-1 at different dosage.

**Figure 4 molecules-28-00513-f004:**
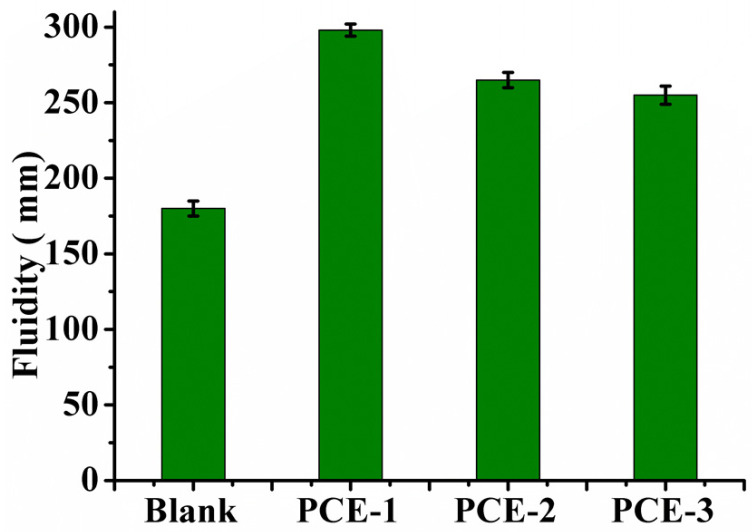
Initial fluidity of β-HH plaster admixed with PCE-1, PCE-2, and PCE-3, respectively.

**Figure 5 molecules-28-00513-f005:**
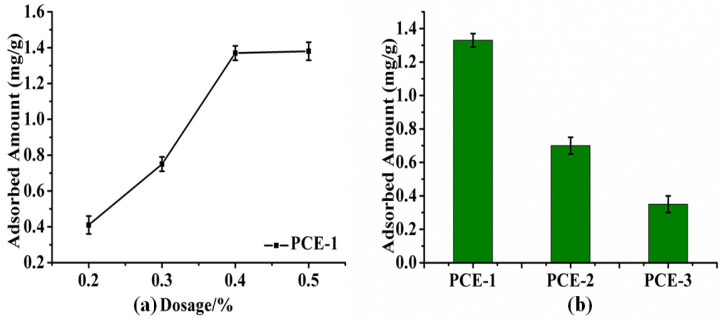
(**a**) Adsorption curve of β-HH with PCE-1 at different dosages; (**b**) adsorption amount of PCEs on β-HH(W/H = 10).

**Figure 6 molecules-28-00513-f006:**
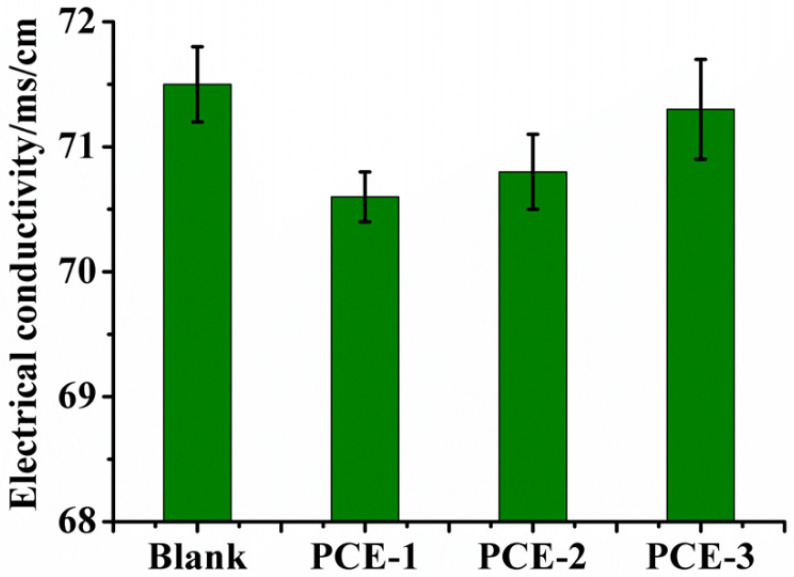
Electrical conductivity of the CaCl_2_ solution with mass concentration of 5% mixed with PCEs.

**Figure 7 molecules-28-00513-f007:**
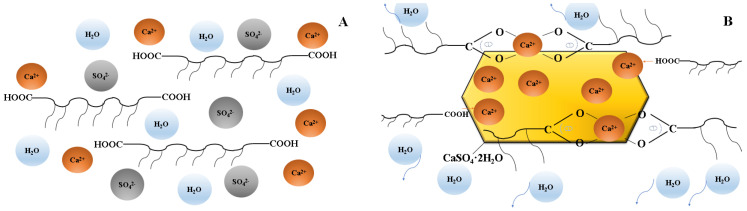
Schematic illustration of coordination between Ca^2+^ and –COO– groups. (**A**) PCEs dissolved in β-HH; (**B**) PCEs adsorbed on the surface of β-HH particles.

**Figure 8 molecules-28-00513-f008:**
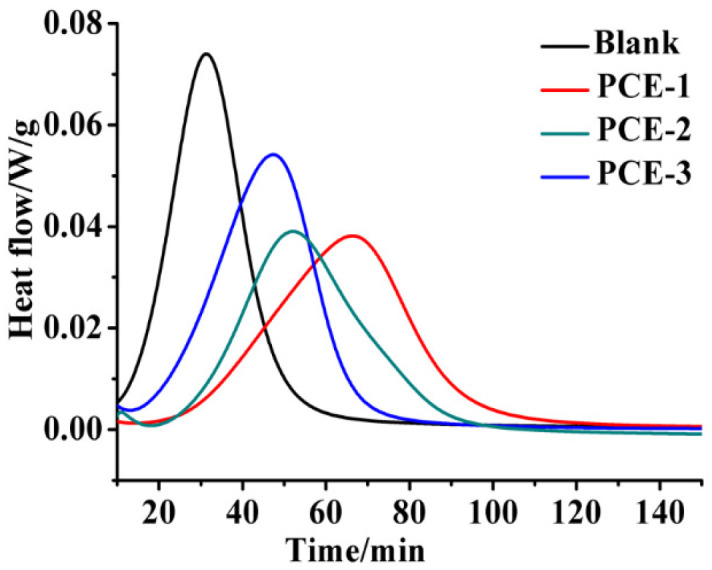
Differential heat flow of β-HH plaster in the absence and presence of PCEs.

**Figure 9 molecules-28-00513-f009:**
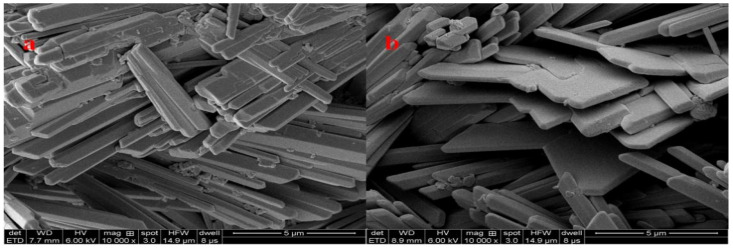
SEM images of the hardened gypsum. (**a**) Blank; (**b**) with PCE-1.

**Table 1 molecules-28-00513-t001:** Molecular weight of as-synthesized PtBA, PAA-PTPEG, and PAA-PTPEG-PAA.

Sample	Theoretical-Mn (g/mol)	Mn (g/mol)	Mw (g/mol)
PtBA	8421	11,900	12,700
PAA-TPEG	42,415	40,400	42,500
PAA-TPEG-PAA	45,375	44,800	49,000

**Table 2 molecules-28-00513-t002:** Compressive and flexural strength of samples with PCEs.

Sample	Flexural Strength (Mpa)	Compressive Strength (Mpa)
Blank	2.9	7.0
PCE-1	4.3	10.0
PCE-2	3.8	9.6
PCE-3	3.4	9.4

**Table 3 molecules-28-00513-t003:** Monomer composition for the A_a_BA_b_-type PCEs synthesized in this work.

Sample	tBA:TPEG	a:b	tBA/TPEG/CTA/AIBN Mole Ratio
PCE-1	5:1	1:1	5:1:0.061:0.06
PCE-2	5:1	2:3	5:1:0.061:0.06
PCE-3	5:1	4:1	5:1:0.061:0.06

## Data Availability

The data presented in this study are available in the article.

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
