# Peer review of "Synthesis of Triblock Polycarboxylate Superplasticizers with Well-Defined Structure and Its Dispersing Performance in β-Hemihydrate Gypsum"

_molecules, 2023, doi:10.3390/molecules28020513_

Round 1

Reviewer 1 Report

The paper will be ready for publication after major revision according to the attached pdf.

Reviewer 2 Report

The paper’s topic is using polycarboxylate superplasticizers to modify building gypsum. The problem is interesting. The fluidizing admixtures are commonly used in cement composites, like cement concrete, but there is little reported research about their use in other building materials, including gypsum. The experiment was, in general, adequately designed and conducted, and the discussion and conclusions presented in the paper are well justified.

The Authors are asked to complete the information about the tested gypsum material – if it was a mortar, the content of the sand (or other filler) is needed; if no sand was used, it was not the mortar, just the gypsum paste. Also, the effect of using superplasticizers in gypsum mix should be more precisely characterized. What was the lowered water-to-HH ratio?

The statement from the Introduction that: ”PCs are generally random copolymer synthesized” is somewhat debatable and needs support by suitable citation.

With the above minor amendments, the paper can be published in the journal Molecules.

Round 2

Reviewer 1 Report

Accept in present form.